# Bi-Allelic Mutations in Zebrafish *pank2* Gene Lead to Testicular Atrophy and Perturbed Behavior without Signs of Neurodegeneration

**DOI:** 10.3390/ijms232112914

**Published:** 2022-10-26

**Authors:** Luca Mignani, Daniela Zizioli, Deepak Khatri, Nicola Facchinello, Marco Schiavone, Giuseppe De Palma, Dario Finazzi

**Affiliations:** 1Department of Molecular and Translational Medicine, University of Brescia, Viale Europa, 11, 25123 Brescia, Italy; 2Faircraft, 94000 Creteil, France; 3Department of Molecular Medicine, University of Padua, 5122 Padova, Italy; 4Department of Medical and Surgical Specialities, University of Brescia, Viale Europa, 11, 25123 Brescia, Italy; 5Clinical Chemistry Laboratory, ASST Spedali Civili di Brescia, 25123 Brescia, Italy

**Keywords:** PANK2, Coenzyme A, neurodegeneration, zebrafish, iron

## Abstract

Coenzyme A (CoA) is an essential cofactor in all living organisms, being involved in a large number of chemical reactions. Sequence variations in pantothenate kinase 2 (PANK2), the first enzyme of CoA biosynthesis, are found in patients affected by Pantothenate Kinase Associated Neurodegeneration (PKAN), one of the most common forms of neurodegeneration, with brain iron accumulation. Knowledge about the biochemical and molecular features of this disorder has increased a lot in recent years. Nonetheless, the main culprit of the pathology is not well defined, and no treatment option is available yet. In order to contribute to the understanding of this disease and facilitate the search for therapies, we explored the potential of the zebrafish animal model and generated lines carrying biallelic mutations in the *pank2* gene. The phenotypic characterization of *pank2*-mutant embryos revealed anomalies in the development of venous vascular structures and germ cells. Adult fish showed testicular atrophy and altered behavioral response in an anxiety test but no evident signs of neurodegeneration. The study suggests that selected cell and tissue types show a higher vulnerability to *pank2* deficiency in zebrafish. Deciphering the biological basis of this phenomenon could provide relevant clues for better understanding and treating PKAN.

## 1. Introduction

The category of Neurodegeneration with Brain Iron Accumulation (NBIA) includes rare, hereditary disorders sharing a complex neurological phenotype with progressive motor dysfunction and the eponymous sign of iron deposition in the cerebral parenchyma, particularly in the basal ganglia [1]. To date, 15 NBIA subtypes have been genetically defined [2]. Pantothenate Kinase Associated Neurodegeneration (PKAN, OMIM # 234200) is one of the most common forms [3] and accounts for a lifetime risk of 0.24/100,000 [4]. In most cases (classic PKAN), the onset is very early in childhood; the main clinical features are extrapyramidal symptoms, particularly generalized dystonia and dysarthria, rigidity, visual impairment, and cognition dysfunction; disease progression is fast and associated with markedly reduced lifespan. Atypical forms are characterized by later onset, slower progression, and more heterogeneous clinical features, including parkinsonism, neuropsychiatric manifestations, speech abnormalities, and cognitive decline [5]. Brain Magnetic Resonance imaging is essential for the diagnostic evaluation of suspected PKAN cases. The “eye of the tiger” sign, which reflects the accumulation of iron in the Globus Pallidus (GP), is an almost constant feature of the disease [6]. The main neuropathological signs found in genetically confirmed PKAN brains are localized in the GP and consist of iron deposits, significant loss of neural cells, axonal spheroids, and gliosis [7]. This focal vulnerability within the brain parenchyma is largely unexplained as yet.

PKAN is due to loss of function mutations in the pantothenate kinase 2 (PANK2) gene [8], which codes for a mitochondrial enzyme catalyzing the phosphorylation of pantothenate (vitamin B5), the first event in the biosynthesis of Coenzyme A (CoA). On these premises, the reduction of CoA concentration in cells and tissues is assumed as the main biochemical determinant of the pathology; of note, most attempts to measure total CoA level in different PKAN experimental models did not provide clear-cut proof in support of this hypothesis [9]. As observed in mice [10], the presence of other PANK enzymes (namely PANK1a and b and PANK3) could exert a compensatory activity and limit the negative effects of PANK2 deficiency in most cells and tissues. Alternatively, other yet unknown biological mechanisms are at work and make the changes in CoA metabolism extremely cell/tissue-specific or compartmentalized and hence not measurable by the approaches so far applied. For the time being, the main culprit of the pathology is not well defined. Nonetheless, we have an extensive description of the distinct biological perturbations induced by PANK2 deficiency and contributing to the neurodegenerative process. Many studies performed in animal [11,12] and cellular models [13,14,15,16] highlighted the presence of mitochondrial abnormalities and dysfunction, including increased fragmentation, structural swelling and damaged cristae, altered mitochondrial membrane potential, and reduced oxygen consumption; this was constantly associated with significant changes in the cellular oxidative status. The direct connection with CoA biosynthesis was confirmed by the rescue capacity of molecules such as pantethine, CoA, fosmetpantotenate, and 4′-phosphopantetheine [11,17,18,19,20], capable of directly or indirectly bypassing the blockage in pantothenate phosphorylation. Altogether, these studies suggest that the accumulation of impaired mitochondria and ROS adducts plays an early and fundamental role in the pathogenesis of PKAN. This connection is supported further by a recent report documenting the involvement of PANK2 and CoA/Acetyl-CoA availability in a PINK1-dependent signaling cascade controlling mitophagy [21]. Less clear is the connection between PANK2, CoA biosynthesis, and iron accumulation, probably because this feature is rarely reproduced in the available experimental systems. While signs of iron dyshomeostasis were commonly detected in in vitro models of the disease [14,22,23], the presence of iron deposits was observed occasionally in fibroblasts [15] and more recently in γ-aminobutyric acid (GABA)ergic neurons and astrocytes [24] obtained from PKAN induced-pluripotent stem cells (iPSc). Nor signs of iron imbalance or deposits of the metal were found in PKAN animal models, such as drosophila [25] or mice [12,26], indicating a major limit of these models in the investigation of the origin and the role of iron depositions in neurodegeneration, particularly in PKAN. Interestingly, an in-depth analysis of the GP of PANK2-KO mice [20] revealed the focal presence of perturbed iron homeostasis, together with defects in pyruvate dehydrogenase (PDH) and mitochondrial complex I activity. As nicely documented also by Lambrechts et al. [27], these changes could be due to inadequate post-translational 4′-phosphopantetheinylation of proteins, including the mitochondrial acyl carrier protein (mtACP). These data indicate that defective protein lipidation represents an important biochemical connection between altered CoA biosynthesis and the neurodegenerative process. Along this line of evidence, the reduced palmitoylation of transferrin receptor and p62 in PKAN models appears to be linked directly with iron overload and defective mitophagy, respectively [21,28]. It is clear that the knowledge about the biochemical and molecular features of PKAN pathology has increased a lot in recent years. Nonetheless, we still do not have a definitive understanding of the main mechanistic driver of the neurodegenerative process. This could partially explain the limited success of the clinical trials that explored the therapeutic potential of molecules assumed to bypass the PANK2-dependent blockage in the CoA biosynthetic pathway [29,30].

To improve the understanding of this rare disease, we have explored the potential of the zebrafish animal model, which combines the versatility of invertebrates with the anatomical similarity of mammals [31], is suitable for genetic manipulation, and provides a valuable platform to perform high-throughput screens of molecules with rescue potential. Firstly, we studied the role of Pank2 in embryonal development by the morpholino approach. Embryos with *pank2* downregulation showed defects in neuronal development, together with perturbations in the formation of the vasculature [32]; interestingly, the overexpression of mutant forms of the human PANK2 in zebrafish embryos induced similar vascular abnormalities [33] and silencing of the enzyme in human umbilical vein endothelial cells resulted in defective angiogenic properties [34]. To extend the study to later developmental and adult stages and generate a stable model amenable to drug screening, we applied the CRISPR/Cas9 technology to generate zebrafish lines carrying biallelic mutations in the *pank2* gene. We obtained a line with the homozygous deletion of 4 bp in exon 1 of the gene, leading to the abrogation of Pank2 expression. The phenotypic characterization of mutant embryos revealed anomalies in the formation of the Caudal Venous Plexus (CVP) and in the development of germinal cells. Adult fish showed testicular atrophy and altered behavioral response in an anxiety test but no evident sign of neurodegeneration.

## 2. Results

### 2.1. Generation and Validation of the pank2 Mutant Line

We designed a guide RNA complementary to a sequence in *pank2* exon 1 (5′-GGGGAAGTGCGCTCAGAGAGCGG-3′) by the CHOPCHOP algorithm (https://chopchop.cbu.uib.no, accessed on 1 March 2015) and synthesized it according to the method described by Gagnon et al. [35]. We injected 50 ng of the sgRNA and 340 ng of the Cas9 mRNA in one-cell stage embryos to obtain the founder chimeric fish (F0 generation). Fish carrying homozygous mutations were generated by standard breeding schemes; we selected a line with a biallelic disruptive deletion of 4 bp in the *pank2* coding sequence for all further studies (*pank2*^ubx1/ubx1^ line in the ZFIN database https://zfin.org/ZDB-ALT-220930-1, hereafter D4). This should result in a frame-shift in the coding sequence and lead to a premature stop codon at amino acid 64, located upstream of the catalytic domain (Appendix A). The analysis of the Pank2 protein level in mutant zebrafish embryos at 48 hpf by Western blotting did not reveal any band, confirming the disruptive nature of the mutation (Figure 1A). The mRNA for the *pank2* gene was reduced by about 50% in mutant embryos as compared with control ones, suggesting the activation of the mRNA decay process (Figure 1B). In contrast with the phenotype observed in embryos injected with *pank2*-specific morpholinos, D4 mutant embryos displayed no evident morphological abnormalities (Figure 2) until 120 days post fertilization (dpf). We detected no difference in the size of the eye diameter or in the body length between mutant and control embryos at 72 hpf. No difference in size was present in adult fish at 1 year of age (Appendix A). To confirm further the disruptive effect of the selected mutation, we injected the previously described *pank2*-specific morpholino [36] in wild-type and mutant embryos and observed their morphology and development at 48 hpf (Appendix A). The downregulation of *pank2* expression induced the expected phenotype in wild-type fish, with about 80% of the embryos showing either severe or mild perturbations of the structure of the caudal plexus, with hemorrhages evident in more than 30% of the injected embryos. On the contrary, D4 embryos exposed to the same morpholino dose were largely normal, with only about 30% of the embryos showing very subtle abnormalities, probably associated with the microinjection procedure, and found in a similar percentage of control embryos. Altogether, these results confirm both the specificity of the phenotype associated with the injection of the morpholino and, above all, the abrogation of *pank2* translation in the D4 mutant line.

The absence of the expected morphological phenotype spurred us to investigate changes in the expression of enzymes involved in CoA biosynthesis, and in particular of other *pank* isoforms (*pank1a* and *pank1b*) and *coasy*. Real-time qPCR analysis evidenced no significant changes in *pank1a* and *pank1b* levels, while *coasy* mRNA amount showed high variability but a significant increase in the mutant embryos (Figure 1B). Measurement of total CoA by a commercial assay showed only a trend for a reduction in the amount of this metabolite in D4 embryos at 48 hpf (Figure 1C).

### 2.2. Evaluation of Mitochondrial Respiration and Cell Death

Defects in PANK2 activity have been associated with perturbations of mitochondrial functioning [11,12]. We evaluated mitochondrial health by measuring the oxygen consumption rate. The graphs in Figure 3 show that the parameters of mitochondrial respiration assessed in basal condition or by exposing embryos to oligomycin and carbonyl-cyanide-p-trifluoromethoxy-phenyl-hydrazone (FCCP) had similar values in wild-type and mutant embryos at 48 and 72 hpf. A minor significant shift was present upon treatment with antimycin A and rotenone in mutant embryos at 48 hpf but not at 72 hpf, suggesting a possible transient perturbation of complex I and/or III in the electron transport chain. We also analyzed the presence of cell death in the embryos at 72 hpf by acridine orange staining and found a similar number of fluorescent dots in the head of wild-type and D4 embryos (Figure 3C). Altogether, these results suggest that the absence of *pank2* function does not affect mitochondrial activity and does not increase the incidence of cell death at the early stages of embryonal development.

### 2.3. Assessment of Neural Development in the pank2 Mutant Line

Even though D4 embryos did not show evident alterations of brain development until 5 dpf, we decided to investigate neuronal development at a cellular level and crossed D4 homozygous mutant fish with transgenic animals expressing GFP under the neuronal differentiation 1 (*neurod1*) gene promoter [Tg(*neurod1*:EGFP)^ia50^]. *neurod1* is a proneural basic helix-loop-helix transcription factor expressed in a large set of postmitotic neurons during embryonal neurogenesis and differentiation [37,38]. We compared the GFP-dependent fluorescence in control and mutant transgenic fish from 24 hpf until 5 dpf without observing changes in the distribution or in the intensity of the signal (Figure 4A). The staining for the pan-neuronal marker Elavl3+4 performed at 48 hpf confirmed this observation (Appendix A).

In PKAN, the neurodegenerative process is largely focused in the GP. Comparative expression analysis of homologous developmental genes suggests that an area in the zebrafish ventral telencephalon could correspond to the pallidum in other vertebrates [39]. To investigate the development of this brain region in more detail, we performed WISH in control and mutant embryos to analyze the pattern of expression of distal-less homeobox 2 (*dlx2a*) mRNA. *dlx* homeobox genes play essential roles in the differentiation, migration, and survival of most GABAergic neurons in the forebrain [40]. The staining decorated the subpallium, the preoptic area, and the prethalamus in control and mutant embryos at 48 hpf without significant differences (Figure 4B).

### 2.4. Vascular Development in pank2 D4 Mutant Embryos

Since *pank2* morphants displayed a defective formation of the CVP, we carefully observed and analyzed the development of this vascular structure in the tail of control and mutant embryos from 24 to 72 hpf. Microscopic examination of live embryos at 30 hpf confirmed the presence of a normal blood flow and seemed to reveal subtle changes in the size of the caudal aorta and the caudal vein. Measurement of the length of these vascular structures normalized to the length of the tail revealed a minor yet significant reduction in D4 mutant embryos (Figure 5). We investigated further the development of the CVP by whole-mount endogenous alkaline phosphatase staining of wild-type and mutant zebrafish embryos at 72 hpf. In the tail of control embryos, the enzymatic reaction revealed the standard honeycomb structure of the primordial CVP, a capillary meshwork with intercapillary spaces and interconnected, lumenized tubules that serves as a conduit for blood circulation. In contrast, in D4 mutant embryos, the formation of the CVP appeared to be perturbed, with the appearance of a single-lumen, dilated caudal vein in place of the tortuous capillary network (Figure 5B). Even though this defect appears to have no long-term consequences in the development of the vasculature structure in D4 mutants, it confirms the previously proposed association of *pank2* activity and CoA biosynthesis with selected processes of the angiogenesis program [32,33,36].

### 2.5. Analysis of Primordial Germ Cells Development in D4 Embryos

Since PANK2-KO mice show oligo/azospermia, we investigated the development of primordial germinal cells (PGCs) in D4 and control embryos at 24 hpf. We performed WISH for the *nanos homolog 3* gene (*nanos3*), expressed in germinal cells to control their proper migration and survival [41]. The staining specifically decorated the PGCs and was clearly less intense in D4 mutants (Figure 6A). Careful microscopic observation of the stained embryos and counting of the *nanos3*-positive cells documented a reduction in the number of PGCs in D4 mutants, from a mean number of 8 in controls to a mean number of 5 in mutant embryos (Figure 6B).

### 2.6. One-Year-Old D4 Mutants Display Testicular Atrophy

To investigate possible long-term effects of pank2 deficiency, we sectioned 1-year-old adult fish and analyzed the structure and size of tissues and organs by hematoxylin-eosin staining. The parallel observation of several longitudinal sections from 5 different 1-year-old control and mutant adult animals revealed no difference in position, structure, and size of most tissues and organs between control and mutant fish (Appendix A). The histological features of the brain, eyes, intestine, liver, and pancreas were superimposable in the two experimental groups. The gonadal tissue in male animals represented an evident exception and was clearly reduced in size in mutant fish (Figure 6C). We quantified this difference by measuring the surface of the tissue in two homologous parasagittal sections from 7 control and mutant fish by Zen 3.5 (Blue Edition, Zeiss). The analysis confirmed the marked testicular atrophy, with a reduction of about 75% of the surface in mutant animals as compared to control ones (Figure 6D). The overall histological structure of the testis appeared to be conserved (Appendix A). We counted the number of spermatozoa in the sperm of 1-year-old fish and found a trend for a reduction in D4 mutants, with no increase in the number of dead cells as compared to controls (Appendix A). Even though we did not evaluate the fertility of male fish by specific experiments, we never had signs of reduced fecundity in the numerous mating procedures performed to obtain D4 embryos. As a confirmation of this observation, when we analyzed histological sections from 3-month-old fish, we found no difference in the size and structure of the gonads of mutant versus control animals (Appendix A). This suggests that the atrophy of the male gonads occurs later in the life of pank2 mutant fish.

We completed the analysis of mutant fish by evaluating the possible accumulation of iron in the brain of 1-year-old fish. We measured the iron content in the brain of 4 wild- type and 5 D4 animals by inductively coupled plasma-mass spectrometry (ICP-MS). The mean values obtained in the two groups were very similar (Appendix A), excluding an accumulation of metal in the whole brain.

### 2.7. Behavioral Studies

We completed the phenotypic analysis of D4 mutant embryos/larvae and adult fish by investigating their motor behavior. Single 5-day-old larvae were placed in wells of a 96-well plate, and their spontaneous movement registered for 120 min according to a well-established scheme (Material and Methods and [42]). The analysis of the total distance swam by the larvae documented a slight but significant reduction for the D4 mutants versus the controls (Appendix A). We then considered the adult fish and performed an open field/novel tank test tracking the movement of single fish from the side of a tank. Single fish were placed in a novel tank with 2 L of water and recorded for 15 min. A novel environment is an anxiogenic stimulus for adult zebrafish, who react by diving to the bottom of the tank to escape potential predators; they explore higher levels of the tank gradually. Anxiolytic drugs attenuate the diving behavior [43,44]. While we did not detect significant differences between control and mutant fish in the total distance moved and velocity (Appendix A), we observed a very different behavior in terms of preference for a specific region of the tank. As clearly shown by the heat map in Figure 7A, when placed in a novel environment, wild-type fish preferred the bottom part of the tank. On the contrary, mutant fish exposed to the same procedure preferred the top part of the tank. This was confirmed by measuring the time spent in three different sections of the tank, bottom, mid and top, during the entire duration of the test (Figure 7B). Wild-type fish spent about 55% of the time in the bottom region and 20% in the top. Both values were about 30% in the case of mutant fish (Figure 7B). When we challenged 3-month-old fish with the same procedure, we found the very same behavior in mutant fish (Figure 7C,D). These analyses suggest that, even in the absence of significant perturbation of the global neuronal development, the deficit of the Pank2 enzyme affects the normal behavior of the zebrafish from the young adult stage onwards. This could be related to the selective alteration of specific neuronal populations and/or neuronal circuitries.

## 3. Discussion

Given the significant level of homology at the genomic level and the similarity in brain structure and functioning, *Danio rerio* represents a valuable experimental setting to investigate the pathogenic mechanism of human neurodevelopmental and neurologic disorders [45]. This became even more so with the advent of easy and affordable genome editing approaches (e.g., CRISPR/Cas9), which allow the site-specific introduction of mutations related to human genetic disorders into the zebrafish genome. Mutations in the PANK2 gene lead to the development of PKAN, a severe and often devastating neurological disorder characterized by the sign of focal iron accumulation in the brain. The limits in the existing experimental models, together with the requirement for platforms to screen for molecules with therapeutic potential, spurred us to develop zebrafish models for PKAN. The partial and transient downregulation of *pank2* mRNA in zebrafish embryos led to abnormalities in the development of neural and vasculature structures that could be rescued by adding pantethine or CoA to the fish water [32]. Now, we have developed a new *pank2* loss of function model, obtained by the CRISPR/Cas9-mediated introduction of a disrupting 4 bp deletion in the coding sequence. To our surprise, the mutant fish did not show overt morphological abnormalities either during the development or at the adult stage. Mutations were inherited with a Mendelian pattern and did not affect survival and life span. The analysis of the oxygen consumption rate indicated normal mitochondrial activity in mutant embryos, and there was no evidence of an increase in cell death. Investigation of neuronal development in the *neurod1* reporter line and by immunostaining for the pan-neuronal marker Elavl3+4 or WISH for *dlx2a*, a transcription factor involved in the specification of GABAergic neurons [46,47], did not reveal changes in the expression level and distribution of the selected neuronal markers. This indicates the normal execution of the neuronal differentiation programs. At the same time, the structure of the brain and the retina in adult mutant fish were superimposable to those of age-matched control animals. The sole morphological anomaly detected during the early stages of development was at the level of the CVP, which appeared as a single, dilated vessel in mutants, versus the tortuous, honeycomb-like structure of wild-type fish. In adult fish, the most striking aberrant anatomical feature observed in mutant fish was the change of the testis size, which was decreased by about 75%. Furthermore, while the motor performances of control and mutant adult fish were comparable, the analysis of the swimming trajectories within three different horizontal layers of the tank (top, medium, and bottom) showed no preference for the mutant fish, a potential indication of a reduced anxiety response to a new environment exposure. Altogether, the phenotype of the *pank2* mutant fish is much milder than that induced by the morpholino approach and devoid of signs of cell death and neurodegeneration either during development or at the adult stage. Since we have evidence that the genome-editing procedure led to the abrogation of *pank2* expression, we infer that in most zebrafish tissues/cell types, the activity of *pank2* is dispensable or that compensatory mechanisms are established to counteract the lack of the enzyme and the potential negative effects upon the biosynthesis of CoA. As observed in mice [48], the activity of *pank1a* and *pank1b*, the other *pank* isoforms in *Danio rerio*, could play a major role from this point of view. We do not have information about their abundance and distribution in zebrafish cells and tissues; their expression levels are not dramatically changed in D4 embryos, so the hypothetical compensatory mechanism does not involve their transcriptional upregulation. In mutant embryos, we detected an increase in *coasy* mRNA level, a potential indication of a feedback loop induced by low CoA levels. Furthermore, a process of transcriptional adaptation elicited by the degradation of the mutant transcripts via the mRNA surveillance pathway could modify the expression of genes not directly involved in the phosphorylation of vitamin B5 but contributing to a compensatory biochemical adjustment [49]. Interestingly, also the microbiome could concur to the maintenance of adequate levels of CoA both in tissues of adult mutant fish and in the yolk of embryos, providing intermediates downstream the first phosphorylation step catalyzed by Pank2 [50].

Mutant embryos did show anomalies in the development of the CVP. The in vivo observation of regular blood flow and heartbeat in mutant fish allows for excluding a hemodynamic origin of this defect. It could instead depend upon specific metabolic features of the venous endothelial cells or changes in the relative expression levels of the different *pank* isoforms, making these cell types more vulnerable to genetic defects. Interestingly, zebrafish embryos treated with drugs interfering with hydroxymethylglutaryl-CoA reductase (HMGCR) (aplexone and simvastatin) showed very similar effects on the development of the CVP and the angiogenesis from endothelial cells of venous origin [51]. In this study, the authors showed that the final effect of aplexone could be partially due to defective C-terminal prenylation of proteins involved in the control of cell migration. For yet unknown reasons, the germ and the venous endothelial cells were most sensitive to HMGCR inhibition, which resulted in defective cell migration. In our mutant embryos, a putative shortage of CoA could limit the HMGCR activity, particularly in these specific cell types, and lead to the observed vascular phenotype. In addition, the reduced number of PGC counted in *pank2* mutant zebrafish embryos could be in line with this observation. Changes in protein’s post-translational modifications directly or indirectly depending on CoA levels and availability appear to be a common sign in different models of NBIA disorders, particularly in those associated with inborn errors of CoA biosynthesis. Drosophila larvae with reduced expression of pank2/fumble showed defective protein acetylation [52], fibroblasts from NBIA patients had low levels of transferrin receptor pamlytoylation [53], and mice and flies with deletion of pank2 showed defective protein 4-phosphopantetheinylation [20,27]. Treatment of cells and animals with substrates bypassing the blockage in CoA biosynthesis prevented the aberrant phenotypes, thus confirming the direct connection with this metabolic pathway. The geranylgeranylation/farnesylation of selected protein could represent another path affected by pank2 malfunctioning. Changes in isoprenoid production would also hinder the synthesis of cholesterol that plays a relevant role in several biological pathways, including the control of membrane fluidity and structure and membrane trafficking. Interestingly, Leoni et al. [54] found reduced levels of cholesterol precursors, lanosterol, and lathosterol in PKAN patients’ serum and fibroblasts. A cell-specific defect in cholesterol production could thus concur with the insurgence of this transient vascular phenotype in *pank2*-mutant embryos.

The reduction of the testis size is the single evident anatomical defect we observed in adult fish of 1 year of age. This is in line with results in mice, where deletion of the *Pank2* gene led to a significant reduction in the weight and size of the male gonad from 6 weeks of age on. It was associated with spatially disordered spermatogenesis, with a possible arrest at the spermatid stage, and with azoospermia. Even though we did not perform specific experiments to monitor the fertility of mutant fish, we never experienced large differences in the number of fertilized embryos collected from breedings of control or mutant fish. This could largely depend on the external fecundation of zebrafish oocytes. These data, as well as the reduced fertility documented in pank2/fumble deficient flies [55] and in *Pank2* KO mice [26], confirm the relevant role played by PANK2 in the spermatogenesis process. Investigating the biological underpinnings of this phenomenon could provide important hints about the vulnerability of selected cell types observed in PKAN disorder.

Finally, adult D4 mutant fish showed reduced anxiety levels when exposed to a novel environment. The phenotype was observed at different adult stages and in animals from different breedings; this should support the specific relationship with the mutation in *pank2*, even in the absence of molecular analysis investigating possible off-target effects of the genome editing approach. It is difficult to speculate about the possible mechanisms connecting *pank2* deficiency with this behavior. It is interesting to observe that data in the literature suggest that, among others, the GABAergic inhibitory transmission plays a role in the pathophysiology of anxiety [56]. Indeed, the GP, which is highly affected in patients with PKAN, has a primary role in the execution and refinement of movements but also appears to be involved in emotional behavior [57]. Therefore, it is tempting to suggest that the behavioral phenotype we documented in D4 mutant fish could be an additional indication of the selective vulnerability of specific (neural) cell populations, possibly of the GABAergic type. Even though we did not detect evident signs of increased cell death, we cannot exclude modification in the development of selected neural populations in terms of number, distribution, and connectivity. More specific work is required to investigate further the phenotype and this speculative hypothesis. The molecular mechanisms of fear and anxiety are highly conserved in vertebrates and are present in zebrafish both during development and at the adult stage. For instance, cortisol levels increase after visual contact with a predator and are reduced by treatment with anxiolytic drugs in the novel tank test [58]. Once the specificity and the “bona fide” features of this behavioral phenotype are confirmed, it should be possible to set up an assay amenable to the screening of molecules with rescuing potential.

In conclusion, our animal model indicates that *pank2* deficiency does not alter the development of zebrafish embryos that reach adulthood and survive without major differences in comparison with control fish. Even though the activity of Pank2 seems to be largely dispensable in zebrafish, selected cell types and tissues show transient or persistent abnormalities, thus suggesting a specific vulnerability to the genetic and biochemical defect. The study of the biological basis of this feature is a big challenge, but it could provide relevant clues for better understanding and treating PKAN disease.

## 4. Materials and Methods

### 4.1. Fish Maintenance

All the fish are stored in specific aquatic habitats developed by TECNIPLAST, these systems are composed of 3/5 L tanks, and each contains more or less 10–15 fish. Water temperature is maintained at 28 °C inside the system with pH ranges between 6 and 8 and conductivity between 500 and 1000 μS. The systems are kept on a Day-and-Night cycle, with 14 h of light and 10 h of dark each day to simulate the fish’s natural habitat. Adult fish are fed three times per day, the juvenile and larvae four times. Food comprises commercially available Gemma Micron (Scretting) and live artemia. All the experiments were performed on embryos before the age of 5 days post-fertilization. Mutant lines were generated following the minister project n° 585/2018. The mating was set up the previous night by organizing the tanks; male and female fish were divided by a transparent Plexiglas separator. Upon fertilization, the eggs were collected using a sieve and placed in a Petri dish containing fish water and incubated at 28 °C. All the experiments were conducted using embryos derived from an incross of double mutant fish for pank2. Control embryos were obtained crossing AB wild-type fish or Tg(*neurod1*:EGFP fish).

### 4.2. Whole-Mount In Situ Hybridization (WISH)

To synthesize the riboprobes for the detection of zebrafish *dlx2a* and *nanos3* transcripts, we amplified specific regions by PCR using as a template the cDNA from the 2-cell stage and oligonucleotide primers. The amplification conditions were the following: 2 min at 95 °C; 30 cycles at 94 °C for 30 s; 60 °C for 30 s, 72 °C for 1 min; followed by a final extension at 72 °C for 5 min. Antisense and sense RNA probes were obtained by in vitro transcription of the cloned PCR product with T7 or T3 RNA polymerase, using a digoxigenin labeling mixture according to the manufacturer’s instructions (Merck KGaA, Darmstadt, Germany). Whole-mount in situ hybridization (WISH) was performed according to a standard method (Thisse and Thisse, 2008). Briefly, embryos and larvae were collected, dechorionated, and incubated at 28 °C at different stages. Embryos were fixed overnight in 4% paraformaldehyde (PFA) at 4 °C, dehydrated through an ascending methanol series, and stored at −20 °C. After treatment with proteinase K (10 μg/mL, Merck KGaA, Darmstadt, Germany ), the embryos were hybridized overnight at 68 °C with DIG-labeled antisense or sense RNA probes (400 ng). Embryos were washed with ascending scale of Hybe Wash/PBS and SSC/PBS, then incubated with anti-DIG antibody conjugated with alkaline phosphatase overnight at 4 °C. The staining was performed with NBT/BCIP (blue staining solution, Merck KGaA, Darmstadt, Germany) alkaline phosphatase substrates. When different types of probes (sense vs. antisense) or fish (injected vs. not-injected) had to be compared, all incubations were carried out at the same time, at the same probe concentration, and, when possible, with the same reagents and solutions.

### 4.3. DNA Extraction from a Single Embryo

DNA extraction from a single embryo was performed following a protocol based on the HotSHOT method. Briefly, the embryo is collected in a sterile Eppendorf tube containing 10 µL of sterile NaOH 50 mM. The sample is boiled at 95 °C for about 12 min and then transferred in ice for 2–3 min; 2 μL of TRIS-HCl pH 7.5 1 M is added to equilibrate the pH.

### 4.4. Extraction of DNA from an Adult Fish

At the age of 8 weeks, it is possible to perform tail cutting for DNA extraction and genotyping on fish. Fish were anesthetized using a diluted tricaine solution, a small part of the fish’s tail was excised using a scalpel, and the excised part of the tail was transferred to a sterile Eppendorf tube; while the fish was transferred back to the tank, ensuring its vitality. The HotSHOT method was applied for the extraction of DNA from the cells of the tail. Briefly, 20 μL of 50 mM NaOH was added to the Eppendorf tube containing the tail fragment; the sample was boiled for 10 min at 95 °C, cooled in ice for 5 min, and 4 μL of Tris HCl 1 M pH 7.5 was added; DNA concentration was assessed by analysis with the NanoDrop 1000 spectrophotometer.

### 4.5. Heteroduplex Mobility Assay (HMA) Analysis

A 15% non-reducent acrylamide gel was prepared. 10 µL of PCR was loaded, and the run preceded until the right separation of the band in TBE 1X. Staining was performed for 10 min in a solution of 5 µL GelRed to TBE buffer. Visualization of the bands was performed on a trans illuminator. Pictures were taken using different cameras.

### 4.6. Western Blot

Twenty embryos for each condition were manually dechorionated. Proteins were extracted by homogenizing embryos in buffer (200 mM Tris/HCl, 100 mM NaCl, 1 mM EDTA, 0.5% NP-40, 10% Glycerol, 1 mM NaFl and sodium orthovanadate) on ice and quantified by a standard bicinchoninic acid method. A total of 50 μg of total protein extract were then separated on 10% polyacrylamide gels and transferred onto membranes (GE AmershamTM HybondTM P 0.45 PVDF). Membranes were cut at the level of the molecular weight of interest, blocked with 2% skim milk in TBST, and incubated with the primary antibody diluted in 2% skim milk in TBST (Abcam ab128298 polyclonal rabbit anti-PANK2 antibody, 1:1000, Origene TA890010 monoclonal mouse anti-actin antibody, 1:1000) and then with the appropriate secondary antibody also prepared in 2% skim milk (1:2000, from Pierce). Images were acquired with Li-Cor Odyssey Image station, and band intensity was quantified by ImageJ software without any modification of the original data. Original images were partially cropped to fit the final figure.

### 4.7. RNA-Extraction and qPCR

Total RNA was extracted from 30 embryos for each different developmental stage and analyzed using TRI-Reagent (Merck KGaA, Darmstadt, Germany) according to the manufacturer’s protocol. RNA was quantified using the My Spect spectrophotometer (VWR International, Radnor, PA, USA). Then, 1.5 μg of total RNA was retro-transcribed to cDNA using Im-Prom reverse transcriptase (Promega, Madison, WI, USA) and oligo(dT) primers following the manufacturer’s protocol. Primers were designed by the Real-Time PCR Tool from IDT. Real-time PCR was performed using the Applied Biosystems^®^ ViiA™ 7 Real-Time PCR System. Reactions were performed in a 10 μL volume, with 100 μM of primer, 5 μL of Syber Green Master Mix (Biorad, Hercules, CA, USA), and 75 ng of cDNA. The amplification profile consisted of a denaturation program (95 °C for 1 min) and 40 cycles of 2 steps amplification (95 °C for 15 s and 60 °C for 30 s) followed by a melting cycle. Each reaction was performed in triplicate. Relative levels of expression were calculated by the ΔΔCT method. All primers are listed in Appendix A.

### 4.8. Coenzyme A Quantification

Coenzyme A quantification was performed using a commercial Coenzyme A Assay Kit (Fluorometric–Green, Abcam ab138889). This kit provides a sensitive fluorimetric assay to quantify CoA by detecting the –SH group. Tissue was listed in a proper volume of assay buffer (provided in the kit) by 3 cycles of vortex for 2 min and 10 min in ice. After complete lysis, samples were measured following the manufacturer’s protocol, and a standard curve of CoA 1, 3, 5, 10, 15, and 30 µM was performed to generate a standard curve. The samples were diluted 1:10 in assay buffer before quantification. Fluorescence intensity was measured at Ex/Em 490/520 nm in the EnSight Multimode Plate Reader (Perkin-Elmer, Waltham, MA, USA). Data obtained were analyzed using Excel software (Microsoft) and GraphPad Prism 8 (Dotmatics, Boston, MA, USA).

### 4.9. Microinjection

The embryos were collected, washed, and prepared on the side of a histological glass. Needles were prepared from borosilicate glass capillaries using a microelectrode puller (Narishige Japane, Puller model: PN-30). Using a microinjector (micromanipulator M275, microinjector Inject Man N12, FemtoJet Eppendorf, Hamburg, Germany) connected with a microscope (Leica, Wetzlar, Germany). After 4 h, perfectly injected healthy embryos were collected.

### 4.10. Venous Return Length Quantification

To measure the venous return length, a short video of about 10 s was acquired using a stereo microscope Axio Zoom. V16 stereo zoom from Zeiss, equipped with Axiocam 506 color digital camera with ZEN Pro software from Zeiss. The length of the caudal vein was measured, starting from the yolk extension end to the last point of the caudal vein before joining with the arterial circulation, by watching the blood flow. Tail length was measured as the distance from the yolk extension end to the tail. Each caudal vein measurement was normalized by its tail length and expressed as a percentage.

### 4.11. Phosphatase Assay

On day 3 of development, embryos were fixed in 4% PFA for 2 h at room temperature and stained for endogenous alkaline phosphatase activity. Embryos were washed two times in PBS and dehydrated by immersing in 25, 50, 75, and 100% methanol in PBT to permeabilize. Embryos were then rehydrated stepwise to 100% PBT. For staining, embryos were equilibrated in Tris buffer (0.1 M Tris HCl pH 9.5; 50 mM MgCl; 0.1 M NaCl; 0.1% tween 20) at room temperature. Once the embryos were equilibrated in Tris buffer, BCIP/NBT 1X was added to start the staining. The reaction was stopped by adding PBS. Embryos were then examined on a stereo-microscope (Serbedzija, Flynn, and Willett, 1999).

### 4.12. Oxygen Consumption Rate Measurement

The oxygen consumption rate of each embryo (OCR) was measured by the Seahorse XF-24 extracellular flux analyzer. Zebrafish embryos at 48 or 72 hpf were staged and placed into the wells of an XF-24 capture microplate (1 embryo per well). Capture screens were placed on top of the embryos in order to keep them in place under the measurement area, and each well was filled with 700 μL of fish water. The measurements were performed at 28.5 °C by adding 10 μM oligomycin, 1 and 2 μM FCCP, 1 μM of rotenone and antimycin A mixture after the stabilization of OCR baseline.

### 4.13. Acridine Orange Staining

Embryos at 48 hpf were dechorionated and incubated for 30 min in acridine orange staining solution (10 mg/L). Embryos were then rinsed three times using fish water and quickly imaged using epifluorescent microscopy (Zeiss Axio Zoom.V16 equipped with Zeiss Axiocam 506 color digital camera and processed using Zen 3.5 (blue Version) software from Zeiss (Oberkochen, Germany)).

### 4.14. Immunofluorescence

Immunofluorescence (IF) was performed on overnight fixed embryos at the desiderated stage and dehydrated in methanol. Briefly, fixed embryos were rehydrated through methanol series with PBST. Then, embryos were incubated in Tris buffer (150 mM pH 9.0) at 70 °C for 15 min, rinsed quickly in dH_2_O, and penetrated in acetone at −20 °C for 20 min. After several washes in dH_2_O, embryos were blocked in freshly prepared 10%SS/2%BSA/PBT for 4 h at 4 °C. Incubation with primary Ab in 2%SS/2%BSA/PBT for 1 day at 4 °C. Embryos were washed in PBT and then incubated in secondary Ab diluted in 2%SS/2%BSA/PBT at RT for 3 h. Wash the unbounded secondary Ab with PBST. IF were visualized with an AxioZoom Zeiss microscope. Elavl 3+4 antibody (Genetex 2456 Alton Pkwy, Irvine, CA 92606, USA, GTX128365) 1:200 in blocking buffer. Goat anti-Rabbit IgG (H+L) Secondary Antibody, DyLight™ 488 antibody DyLight 488 (Thermofisher Waltham, Massachusetts, USA, Catalog # 35552) 1:1000.

### 4.15. Histological Section and Haematoxylin-Eosin Staining

Adult zebrafish at 1-year-old and 3 months old were anesthetized by tricaine overdose. Once dead, part of the tail was resected, and a fissure at the abdominal level was performed using scissors. Fish were fixed in Bouin’s solution (picric acid, acetic acid, and formaldehyde) overnight at RT on a shaker. The day after, they were washed several times in ethanol 70% and ammonia until the fish became completely white. Histological section and HE staining were performed by our collaborator in Sezione di Anatomia Patologia, Spedali Civili di Brescia. Images were taken after staining using the AxioZoom Zeiss microscope.

### 4.16. Ejaculate Collection

Ejaculate was collected following a standard protocol. Briefly, fish were isolated from females for 2 days before stripping. Males were anesthetized in Tricaine 1× (0.16 mg/mL), gently dried on a paper towel, and placed in a dampened sponge ventral side up, with their genital papilla exposed. The ventral surface of the fish was further dried to remove any excess water that could prematurely activate the sperm. The fish’s abdomen was gently pushed with soft plastic tweezers, and the ejaculate was collected from the genital papilla by means of a Drummond microdispenser equipped with a 5 μL microcapillary tube. The whole ejaculate was diluted and preserved until analyses (within 1 h) in 30 μL of zebrafish sperm immobilizing solution (ZSI: 140 mM NaCl, 10 mM KCl, 2 mM CaCl_2_, 20 mM HEPES, pH = 8.5).

### 4.17. Sperm Concentration

Sperm counts were performed using an improved Neubauer hemocytometer under 400× magnification after properly diluting 2 μL of a subsample taken from the whole ejaculate maintained in ZSI. The sample was gently mixed with a micropipette before filling the chamber. The average of three counts per sample was used to estimate sperm concentration, considering dilution steps, and expressed as number of spermatozoa/mL.

### 4.18. Sperm Viability

Sperm viability was measured as the proportion of alive sperm. A membrane-permeable nucleic acid stain (SYBR14) labeled live sperm in green, and a membrane-impermeable stain (propidium iodide, PI) labeled dead sperm in red. Two microliters of the sample were diluted by the addition of 10 μL of ZSI, first stained with the addition of 1.5 μL SYBR14, incubated for 5 min at 36 °C and, then stained with the addition of 1 μL of PI, followed by 5 min of incubation at 36 °C. Ten microliters of the stained sample were deposited on a Burker chamber, gently covered with a coverslip, and examined under a fluorescence microscope (Leica M165FC dissecting microscope equipped with a DFC7000T camera, Leica Microsystems) at ×400 magnification. The proportion of live sperm was calculated from three pictures per slide, and the mean values were used for the analyses.

### 4.19. Iron Quantification

Protein extracts from zebrafish brains were digested in 70% nitric acid and deionized water (1:1) for 1 h at 70 °C. Iron concentration was then determined by Inductively Coupled Plasma-Mass spectrometry (ICP-MS) on an ELAN DRC II (PerkinElmer, Waltham, MA, USA) using the analytical technique total quant (begin mass 49 a.m.u., end mass 58 a.m.u.) with external calibration and using the DRC with ammonia (flow 0.7 mL/min, RPq 0.65). The instrument was calibrated using a standard solution (Multielement ICP-MS Calibration Standard 3, Matrix per Volume: 5% HNO_3_ per 100 mL, Perkin Elmer Plus) at a concentration of 10 μg/L. Each sample was analyzed three times.

### 4.20. Larval Stages Behavioral Analysis

A single embryo was collected at 4 hpf in a 96 square well plate with 100 µL of fish water and grew until 5 dpf in the incubator at 28 °C with a light–dark cycle. The plate was placed in the system’s holder. The system was set to acquire for 2 h, 10 min in the darkness and 10 min with a light stimulus, repeated six times. Data were analyzed using Noldus Ethovison software.

### 4.21. Adult Stages Behavioral Analysis

The behavior of adult zebrafish was tested following the protocol for the novel tank test. Each zebrafish was collected from its tank using a net and positioned in a breeding tank filled with fresh water from the Techniplast system. The recording of each sample was performed from the side view of the tank for 15 min using Ethovision Software (Noldus, Wageningen, the Netherlands). Each trial was manually analyzed to avoid mistakes introduced by the detection system. Data were analyzed using Noldus Ethovison software. The virtual spatial division of the tank was performed using Ethovision; each part created had the same perimeter.

### 4.22. Testis Area Analysis

Testis area was calculated by measuring the area of the tissue using ZEN Pro software. From the area, we selected the biggest two and took the mean between them. This operation was done for all the sections of all the fish.

### 4.23. Adult Length Measurement

Adult zebrafish were anesthetized using tricaine, dried using paper tissue, and positioned on graph paper. Images were taken using a camera phone (iPhone 12, Apple) and used to measure the length of each fish.

### 4.24. Microscopy

The images of embryos were obtained with a compound Axio Zoom.V16 stereo zoom microscope from Zeiss, equipped with Axiocam 506 color digital camera with ZEN Pro software from Zeiss. The fluorescence of transgenic lines was visualized using an illuminator HXP 200C from Carl Zeiss. For in vivo analyses, embryos and larvae were anesthetized with tricaine 0.16 mg/mL.

### 4.25. Statistical Analysis

Statistically significant differences between different types of embryos were calculated by Student’s *t*-test, one-way ANOVA with Sidak’s correction for multiple comparisons, or Mann–Whitney U test (GraphPad V8, Dotmatics, Boston, MA, USA), as indicated in each figure legend. *, **, *** and **** indicate *p* < 0.05, <0.01, <0.001 and <0.0001), respectively.

## Figures and Tables

**Figure 1 ijms-23-12914-f001:**
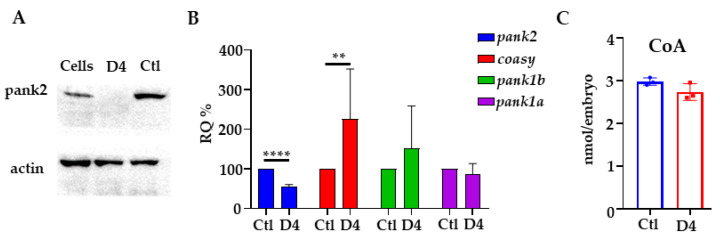
**Molecular characterization of D4 mutant embryos.** (**A**) Western blot for Pank2 in mutant (D4) and control embryos (Ctl). Lysates from SH-SY5Y cells (Cells) were run as controls. (**B**) qPCR analysis on mutant and control embryos for *pank2*, *coasy*, *pank1a,* and *pank1b*. (**C**) Coenzyme A quantification by a fluorimetric assay. Data derived from the analysis of 20 embryos/group, with two biological replicates for Western blotting and at least three for qPCR and CoA quantification. ** *p* < 0.01; **** *p* < 0.0001, Student’s *t*-test.

**Figure 2 ijms-23-12914-f002:**
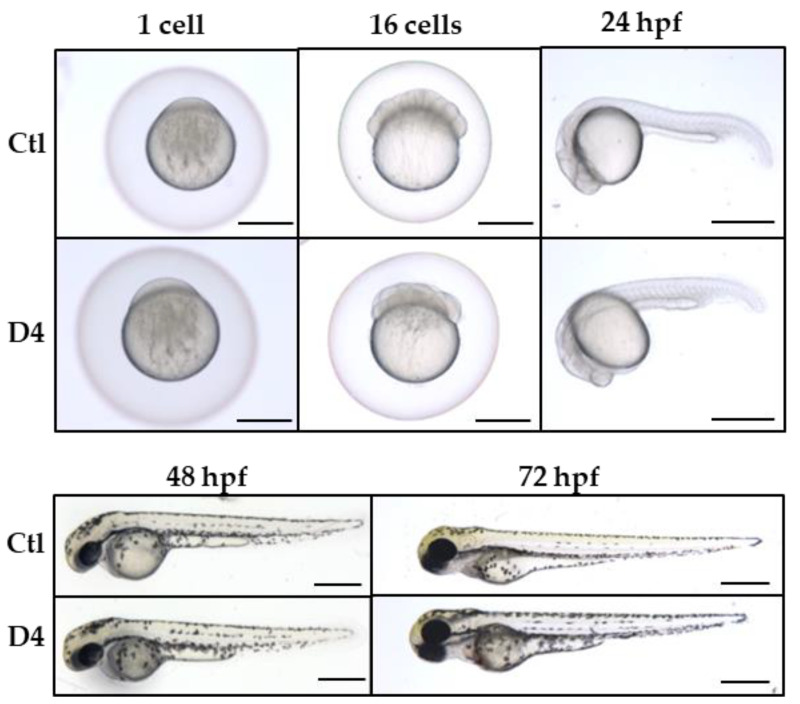
**Morphological characterization of D4 mutant embryos.** Comparison of the morphological features of control and mutant embryos from 1 hpf till 3 dpf. Three biological replicates, N > 100 for Ctl and D4. Size bar = 500 µm.

**Figure 3 ijms-23-12914-f003:**
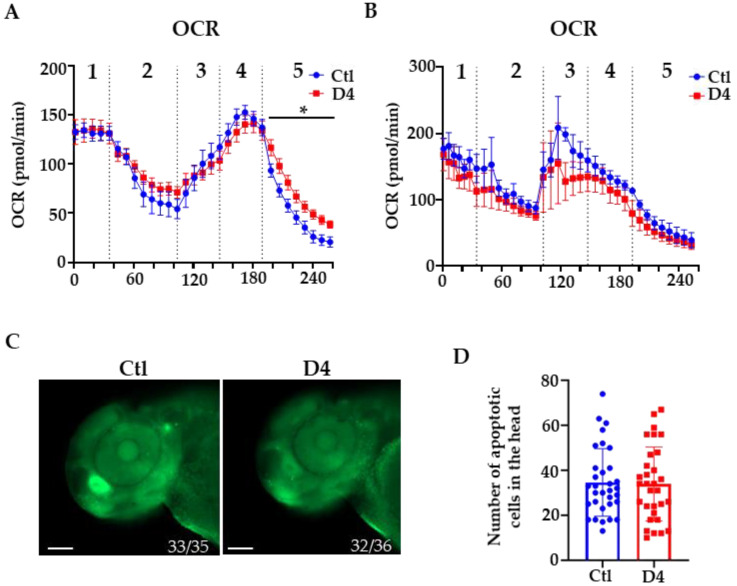
**Oxygen consumption rate (OCR) and cell death analysis**. (**A**) Oxygen consumption rate (OCR) in control (Ctl) and mutant (D4) embryos at 48 hpf (Ctl N = 9; D4 N = 10) and (**B**) at 72 hpf (Ctl N = 9; D4 N = 10). 1 = basal rate; 2 = 10 μM oligomycin, 3 = 1 μM FCCP; 4 = 2 μM FCCP; 5 = 1 μM rotenone and antimycin A. * *p* < 0.05, Student’s *t*-test. (**C**) Lateral view of acridine orange stained embryos at 48 hpf. The numbers at the bottom of the images indicate the ratio between embryos with the observed phenotype and the total number of stained embryos. Size bar = 100 µm. (**D**) Quantification of acridine orange positive cells in the brain region (Ctl N = 33, D4 N = 32; two biological replicates).

**Figure 4 ijms-23-12914-f004:**
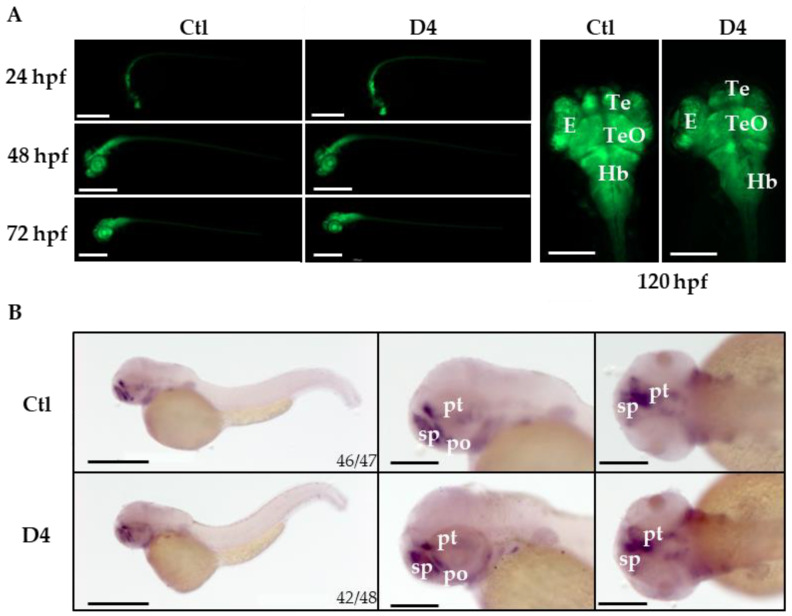
**Neuronal characterization of control and mutant embryos**. (**A**) Representative images of control and mutant Tg(*neurod1*:EGFP) embryos at different stages of development. Size bar = 500 and 200 µm. The analysis was performed in 3 experiments, and more than 100 embryos were analyzed in total per group. (**B**) Whole-mount in situ hybridization for *dlx2a* performed in embryos at 48 hpf. The experiment was performed twice. The numbers at the bottom of the images indicate the ratio between embryos with the observed phenotype and the total number of stained embryos. Size bar = 500 and 200 µm. E, eye; Hb, hindbrain; Ob, olfactory bulb; Te, telencephalon; TeO, tectum opticum; pt, prethalamus; po, preoptic area; sp, subpallium.

**Figure 5 ijms-23-12914-f005:**
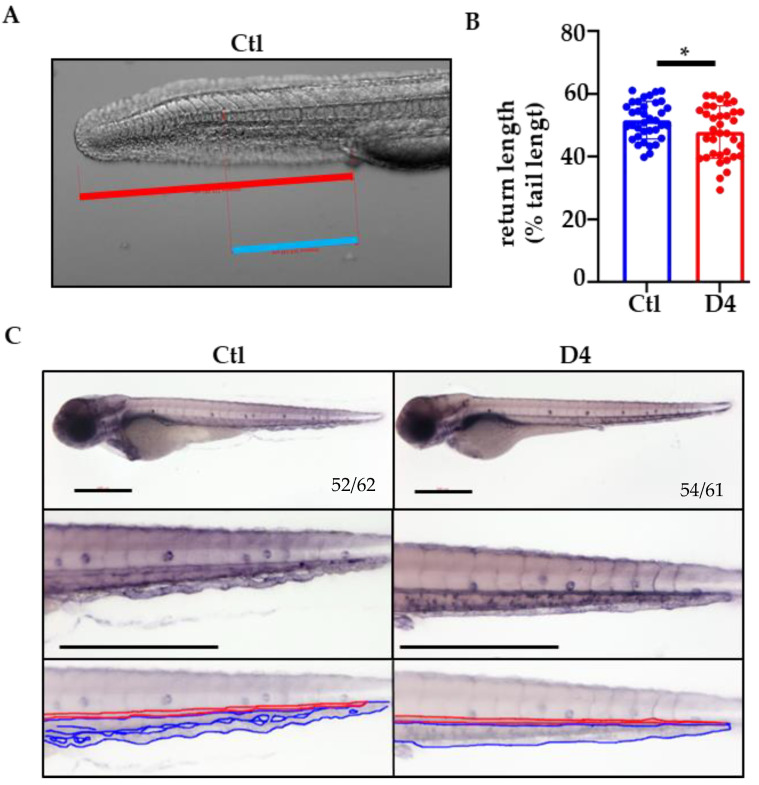
**Effects of the absence of *pank2* on vasculature development.** (**A**) Measurement of the venous return in embryos at 24 hpf; the blue line indicates the length of the caudal vein, and the red line is that of the tail. The measurement was expressed as the ratio between the two values (percentage). (**B**) The graph in the panel shows the mean values obtained in control and mutant embryos (Ctl = 51.6%; D4 = 47.9%). The experiment was repeated twice on 36 embryos/group. * *p* < 0.05, Student’s *t*-test. (**C**) Representative images of the phosphatase assay for vessel visualization on 72 hpf embryos. The red line decorates the dorsal aorta; the blue one is the caudal plexus. The numbers at the bottom of the images indicate the ratio between embryos with the observed phenotype and the total number of stained embryos. The experiment was performed 3 times on at least 60 embryos/group. Size bar = 500 µm.

**Figure 6 ijms-23-12914-f006:**
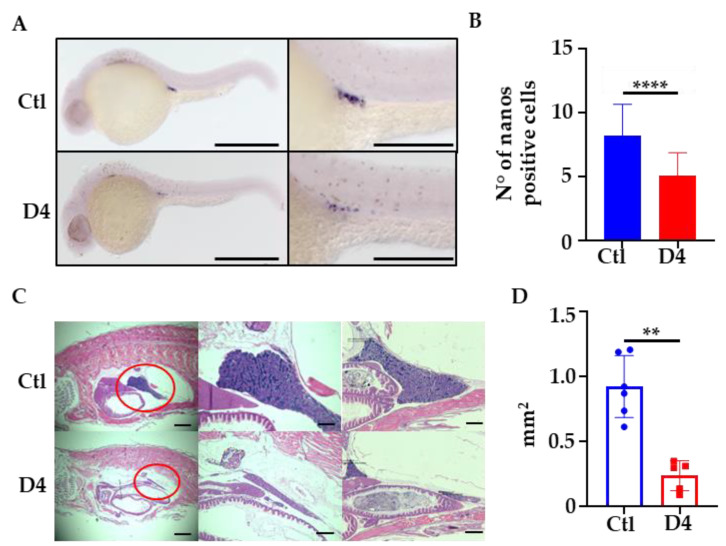
**Primordial germ line and testis evaluation.** (**A**) WISH for *nanos3* in 24 hpf embryos. Size bar = 500 µm and 200 µm. (**B**) Quantification of *nanos3* positive cells in control (Ctl) and D4 embryos (Ctl mean = 8.18 N = 83; D4 mean = 5.07 N = 85, **** *p* < 0.0001, Student’s *t*-test). (**C**) Representative histological sagittal sections of 1-year-old fish, stained with hematoxylin-eosin. Red circle indicates the testis. N = 5 for each group. Magnification 10× and 40×. Size bar = 2000 µm and 500 µm (**D**) Quantification of testis area performed on histological sections; Ctl mean = 0.92 mm^2^; D4 mean = 0.24 mm^2^ ** *p* < 0.01, Mann–Whitney U test.

**Figure 7 ijms-23-12914-f007:**
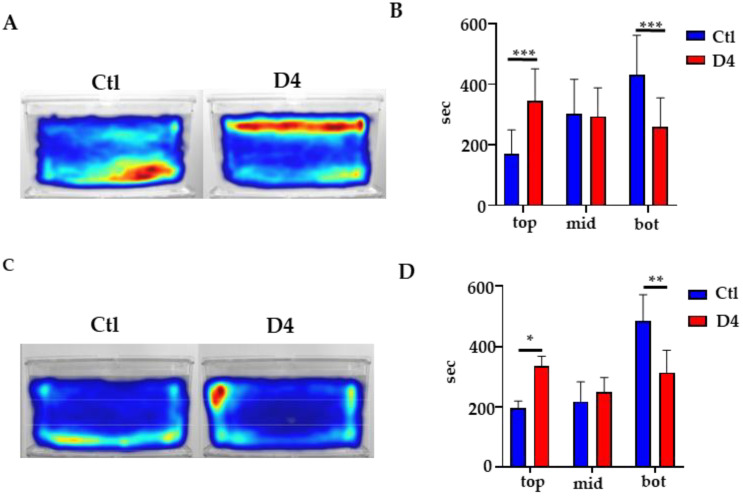
**Behavioral analysis in adult fish.** (**A**) Heat map visualization of the cumulative track of 1-year-old fish obtained with Noldus EthoVision software. (**B**) Quantification of the time spent in the top (top), middle (mid), and bottom (bot) part of the tank by 1-year-old fish; N = 11 for Ctl and 12 for D4 *** *p* < 0.001, ANOVA with Sidak’s correction for multiple comparisons. (**C**) Heat map visualization of the cumulative track of 3-month-old fish obtained with Noldus EthoVision software. (**D**) Quantification of the time spent in different sections of the tank (top, mid, and bot) by 3-month-old fish; N = 3, * *p* < 0.05, ** *p* < 0.01, ANOVA with Sidak’s correction for multiple comparisons.

## Data Availability

The data presented in this study are available on request from the corresponding author.

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
