# Peer review of "Bi-Allelic Mutations in Zebrafish pank2 Gene Lead to Testicular Atrophy and Perturbed Behavior without Signs of Neurodegeneration"

_ijms, 2022, doi:10.3390/ijms232112914_

Round 1

Reviewer 1 Report

I have received the research article "Bi-allelic mutations in zebrafish pank2 gene lead to testicular atrophy and perturbed behaviour without signs of neurodegeneration" by Luca Mignani et al. for evaluation.

In their study, the authors have used a genetic model (biallelic mutations in pank2 gene) of the zebrafish animal model to explore the pathology of common forms of neurodegeneration with brain iron accumulation. The study concludes that pank2 deficiency in adult zebrafish showed testicular atrophy and altered behavioral response in an anxiety test, but no evident signs of neurodegeneration.

The work is timely and important in building on growing evidence to decipher the biological basis of this phenomenon and could provide relevant clues for better understanding and treating Pantothenate Kinase Associated Neurodegeneration (PKAN). The study is novel and experiments are well planned and executed nicely. However, I have some critical comments about this study. My specific comments are mentioned below:-

Major comments:

1-    The pank2 gene is critical for proper mitochondrial functioning. I would suggest testing mitochondrial activity in pank2 deficient zebrafish.  

2-    Authors should also check the level of iron in Globus Pallidusin the brain of pank2 deficient zebrafish.

3-    Investigators have not used any quantitative markers of neurodegeneration as a confirmatory experiment for PKAN.

Minor comments:

1-    Authors should provide the proper justification why the level of Coenzyme A is not changed in pank2 deficient zebrafish while the pank2 gene is critical for Coenzyme A.

2-    In figure 1 C authors should plot the bar diagram showing all the data points (replicates) as presented in figure 4 B.

3-    Authors should also provide the same formate of bar diagram for figure 5 B and D.

4-    Parametric statistical tests (Student’s T-test) are not acceptable for a small sample (especially; n=5). I would recommend adopting non-parametric statistical tests.

Reviewer 2 Report

Comments

In this manuscript the authors have studied the role of pank2 mutation in neurodegeneration

The manuscript raises several questions

1.      First and foremost, I couldn’t  access the supplementary figures and tables

2.      In figure 1, In the D4 mutant model, authors state that the biallelic deletion of 4bp causes a frameshift mutation and an introduction of a stop codon, if so then how is any mRNA for the pank2 gene being synthesized? Is the mRNA obtained for pank2 truncated?

3.      Is the regulation of pank2 post-transcriptional?

4.      Can the authors comment on the role of other compensatory mechanisms in D4 mutants that could ensure the normal functioning of the neural differentiation programs? What happens to pank1a, 1b and pank3 isoforms? The authors show that the levels of pank1a and pank1b is unchanged most deletion of pank2. What does this mean?

5.      Is the lack of morphological and anatomical defects on the D4 mutants related to a species difference since the authors point out that other species like mice and humans show dramatic alterations not limited to just neurological alterations.

6.       What happens to the mitochondrial morphology in the D4 mutant embryos? Are there any changes in the cristae, fragmentation or swelling ? How about oxygen consumption rate?

7.      How does this study compare w the other studies in the field that have shown that deficiency of pank2 leads to neurodegeneration? How do the authors reconcile these differences

8.      Do authors find any iron deposition in their model of the zebra fish? The authors have repeatedly pointed out this is a feature not observed in other models like that of mice and drosophila. Has it been shown for other zebra fish models?

Round 2

Reviewer 1 Report

Accepted

Reviewer 2 Report

The authors have adequately addressed the points raised by me.